# Photo-induced halide redistribution in organic–inorganic perovskite films

Dane W. deQuilettes[1], Wei Zhang[2,†], Victor M. Burlakov[2,3], Daniel J. Graham[4], Tomas Leijtens[2], Anna Osherov[5], Vladimir Bulović[5], Henry J. Snaith[2], David S. Ginger[1] & Samuel D. Stranks[5,6]

Organic–inorganic perovskites such as $CH_3NH_3PbI_3$ are promising materials for a variety of optoelectronic applications, with certified power conversion efficiencies in solar cells already exceeding 21%. Nevertheless, state-of-the-art films still contain performance-limiting non-radiative recombination sites and exhibit a range of complex dynamic phenomena under illumination that remain poorly understood. Here we use a unique combination of confocal photoluminescence (PL) microscopy and chemical imaging to correlate the local changes in photophysics with composition in $CH_3NH_3PbI_3$ films under illumination. We demonstrate that the photo-induced 'brightening' of the perovskite PL can be attributed to an order-of-magnitude reduction in trap state density. By imaging the same regions with time-of-flight secondary-ion-mass spectrometry, we correlate this photobrightening with a net migration of iodine. Our work provides visual evidence for photo-induced halide migration in triiodide perovskites and reveals the complex interplay between charge carrier populations, electronic traps and mobile halides that collectively impact optoelectronic performance.

[1] Department of Chemistry, University of Washington, Box 351700, Seattle, Washington 98195-1700, USA. [2] Department of Physics, University of Oxford, Parks Road, Oxford OX1 3PU, UK. [3] Mathematical Institute, OCCAM, Woodstock Road, University of Oxford, Oxford OX2 6GG, UK. [4] Department of Bioengineering, University of Washington, Box 351653, Seattle, Washington 98195-1653, USA. [5] Research Laboratory of Electronics, Massachusetts Institute of Technology, 77 Massachusetts Avenue, Cambridge, Massachusetts 02139, USA. [6] Cavendish Laboratory, J. J. Thomson Avenue, Cambridge CB3 0HE, UK. † Present address: School of Chemistry, University of Lincoln, Beevor Street, Lincoln LN6 7DL, UK. Correspondence and requests for materials should be addressed to S.D.S. (email: sds65@cam.ac.uk).

Organic–inorganic metal halide perovskites such as $CH_3NH_3PbI_3$ are generating a great deal of excitement for their potential applications in a variety of high-performance optoelectronic devices including solar cells, light-emitting diodes, photodetectors and lasers[1,2]. These applications are enabled by the favourable material properties of these perovskites, which include long charge carrier diffusion lengths[3–5], high absorption coefficients with a sharp absorption edge[6] and a remarkably high photoluminescence (PL) quantum efficiency[7,8]. This latter property is particularly important for approaching the highest photovoltaic device performances at the Shockley–Queisser limit, in which all non-radiative recombination is eliminated[9]. Nevertheless, we recently reported that the PL lifetimes and intensities vary substantially between different grains even in high-quality films[10]. These observations are consistent with the presence of trap states that act as non-radiative recombination sites and influence the recombination kinetics[8,10–13]. In addition, the bulk optoelectronic properties such as photoluminescence[8,14], electroluminescence[15,16] and photoconductivity[17] have also been shown to rise slowly over time under illumination or current flow. These observations indicate that, despite their remarkable device efficiencies, the perovskite films are still far from optimized for stabilized optoelectronic device performance.

It has been suggested that ionic migration in these perovskite materials could impact optoelectronic performance and affect device operation[16,18,19]. A substantial amount of recent evidence indicates that the hysteresis effects in current–voltage measurements originate from a migration of mobile ionic species, where positive and/or negative ionic species migrate to opposite electrodes under an internal field and impact charge collection[16,18–23]. Recent poling experiments on architectures with contacts have revealed evidence for the movement of ions under applied fields[24–28], which can sometimes lead to degradation[24,26]. However, there are no reports presenting visual evidence for ionic migration in perovskite films without contacts and the ensuing impact of migration on the optoelectronic properties is not well understood.

Here we use a combination of confocal fluorescence microscopy and chemical imaging through time-of-flight secondary-ion-mass spectrometry (ToF-SIMS) to study high-quality thin films of the neat perovskite $CH_3NH_3PbI_3$ without any contacts. We show that the PL lifetime and intensity increase significantly over time under illumination, and that these changes correspond to an order-of-magnitude reduction in the bulk trap state density. On the microscale, we find that light-soaking preferentially brightens regions with higher trap state densities and also induces a redistribution in local emission intensities. We use ToF-SIMS depth profiling to show that the rises in PL correlate with a redistribution of iodine in the film, thereby providing strong evidence for a photo-induced halide migration effect. Our results highlight the interplay between free carriers, traps and halide migration and their collective impact on optoelectronic performance.

## Results

### Bulk changes in photoluminescence under illumination.
In Fig. 1a, we show bulk time-resolved PL decays of a neat $CH_3NH_3PbI_3$ film prepared on glass and corresponding to excellent device performance[29] (Supplementary Methods), and the time-integrated intensity in Fig. 1b, over a time period of 10 min under pulsed illumination *in vacuo*. Initially, the PL lifetime is very fast with a significant fraction of non-radiative decay. However, over a period of tens to hundreds of seconds under illumination, the lifetime and total intensity both increase

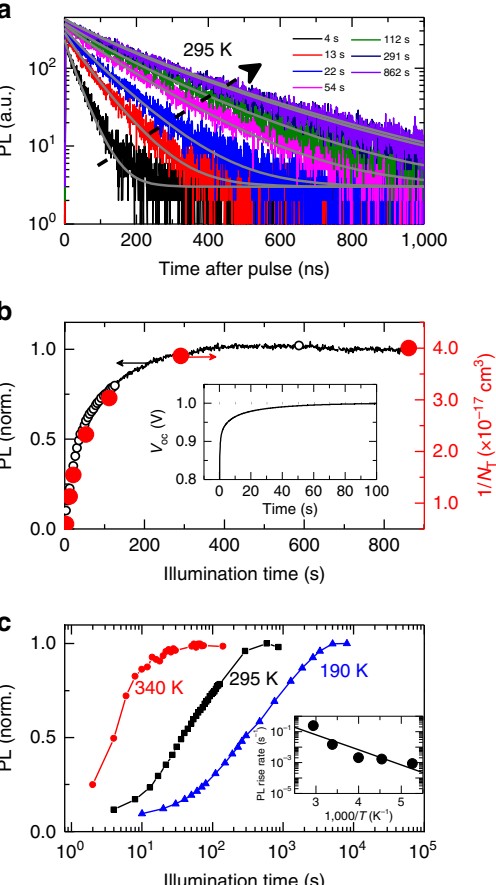

**Figure 1 | Bulk changes in photoluminescence (PL) over time under illumination.** (**a**) A series of time-resolved PL decays from a $CH_3NH_3PbI_3$ film measured over time under illumination. The sample was photoexcited with pulsed excitation (507 nm, 1 MHz repetition rate, 0.3 μJ cm$^{-2}$ per pulse) and the emission was detected at 780 nm. (**b**) The PL over time under initial illumination determined from integrating acquired PL decays (black open symbols) or monitoring the PL count rate (black solid line). The red symbols are inverse trap densities $1/N_T$, where $N_T$ are extracted from the fits to the data in **a** (grey lines). Inset: open-circuit voltage ($V_{oc}$) rise of a full solar cell with an illumination intensity comparable to full sunlight (532-nm cw laser, ~60 mW cm$^{-2}$). (**c**) The normalized integrated PL over time under illumination at different temperatures. Inset: temperature dependence of the rate of PL rise deduced from exponential fits to the data in the main panel (Supplementary Notes 5 and 6). The solid line represents a fit to the data using the Arrhenius relation to extract an activation energy ($E_a = 0.19 \pm 0.05$ eV).

and eventually stabilize after ~10 min (refs 8,14). These observations are consistent with a substantial reduction and stabilization of fast non-radiative decay pathways (herein termed 'cleaning'), and correlate closely with the rise in open-circuit voltage, as we show in the inset of Fig. 1b. The extent of the PL enhancements is influenced by different atmospheres[14,30,31], though the effects are still observed in extremely low oxygen level environments including *in vacuo* and nitrogen and also in films made using other fabrication routes (see Supplementary Fig. 1; Supplementary Note 1), suggesting that the general photo-induced rise behaviour is intrinsic to these polycrystalline films. We note that the PL spectrum shape and position appears unchanged (Supplementary Fig. 2; Supplementary Note 2), suggesting that we are monitoring the same emissive species over the course of the measurement. We also do not observe any significant changes in X-ray diffraction patterns of the films

acquired while light soaking, indicating that the illumination is not inducing large changes in crystal structure or phase composition (Supplementary Fig. 3) (ref. 19).

We have recently developed a model that is able to describe the photoluminescence kinetics in perovskite films in the presence of $N_T$ electronic subgap trap states (Supplementary Fig. 4; Supplementary Note 3) (ref. 8). The fits from the model are shown with the grey lines in Fig. 1a, and the extracted trap densities are shown in Fig. 1b, exhibiting the derived inverse dependence with PL intensity (Supplementary Note 4). The initial trap density of the sample just after illumination ($t = 4$ s) is $\sim 1.7 \times 10^{17}$ cm$^{-3}$ and this slowly reduces to eventually stabilize at $\sim 2.5 \times 10^{16}$ cm$^{-3}$ ($t = 862$ s, total photon dose of 258 J cm$^{-2}$). This order-of-magnitude reduction in trap density under illumination also translates to increased emission and device photovoltage. Importantly, the stabilized values ($\sim 10^{16}$ cm$^{-3}$) are still well above the trap densities reported for single crystals ($\sim 10^{10}$ cm$^{-3}$) (refs 32,33), suggesting that the polycrystalline films still have significant scope for improvement and that light soaking alone may not close the gap in trap state density with their single crystal counterparts. To make the connection between the trap state density and slow photo-induced optoelectronic improvements explicit, we find that the time taken to reach stabilized emission for both cleaved and uncleaved surfaces of single crystal samples is greatly reduced (Supplementary Fig. 1). We note that these results agree with recent theoretical work investigating the effect of different densities of sub-gap states on the open-circuit voltage, where the authors found that the voltage is significantly reduced below its theoretical limit at trap densities of $> 10^{17}$ cm$^{-3}$ but not significantly impacted at trap densities of $\sim 5 \times 10^{15}$ cm$^{-3}$ (ref. 34) These findings are also in agreement with recent reports showing a light-induced increase in electron diffusion length[35] as well as a reduction in the surface potential barrier at the grain boundaries[5].

We show the normalized integrated PL over time at different temperatures in Fig. 1c (see Supplementary Fig. 5, Supplementary Note 5 for decay curves and extracted trap densities at each temperature). At low temperature (190 K), stabilized emission is reached after 10,000 s ($\sim 3,000$ J cm$^{-2}$), whereas at high temperature (340 K), the PL reaches a stable output after just $\sim 100$ s ($\sim 30$ J cm$^{-2}$). This compares with $\sim 1,000$ s ($\sim 300$ J cm$^{-2}$) at room temperature (295 K). We determine an activation energy for the process of $E_a = 0.19 \pm 0.05$ eV using an Arrhenius fit to the rises in PL rates (Fig. 1c inset, see Supplementary Fig. 6, Supplementary Note 6). We also find that the time taken to reach stabilized emission varies dramatically with illumination intensity, with higher intensities leading to substantially faster rise times (Supplementary Fig. 7; Supplementary Note 7). The very slow time scales, strong dependencies on temperature and intensity, and an activation energy on the order of hundreds of milli-electronvolts are consistent with a photo-induced ion migration phenomenon[18,19,36] and will be discussed further below (also see Supplementary Note 8).

**Local changes in photoluminescence under illumination**. We now investigate how microstructure impacts the PL rise behaviour by considering the correlated scanning electron microscopy (SEM) micrograph (Fig. 2a) and photoluminescence image shown in Fig. 2b. We observe a heterogeneous distribution of grains with median grain size of $\sim 0.76$ µm (Supplementary Fig. 8; Supplementary Note 9), where some regions are particularly bright in emission, while other grains and grain boundaries are comparatively dark and correspond to higher trap state densities[10]. In Fig. 2c, we monitor the local emission of different spots over time under continuous-wave (CW)

532-nm illumination at an intensity equivalent to $\sim 3$ suns (188 mW cm$^{-2}$) with a waist ($w$) spot size of $\sim 500$ nm. A bright spot (red triangle) starts at a comparatively high emission level but does not substantially increase over time (relative change by a factor of 1.4). In contrast, a dark spot (blue circle) rises in intensity by nearly an order of magnitude after $\sim 3$ min of illumination (factor of 8.7 relative change). An intermediate spot (green square), also rises by a factor of $\sim 1.6$ over the same period.

These results provide important insight into the bulk measurements presented earlier, which represent an average across the distribution of grains ($w \sim 17$ µm). While the bulk PL increases over time under illumination, each individual grain shows dramatically varied rise time behaviour depending on the initial trap state density. Dark spots rise significantly but the relative change in bright spots is much less, and the net result is an increase in spatially averaged PL eventually reaching a stabilized bulk PL, with the enhancements closely following the spatial profile of the laser (Supplementary Fig. 9; Supplementary Note 10).

To investigate the transient behaviour of the perovskite active layer under solar conditions, we show a fluorescence image in Fig. 3a before exposing the entire film to full-simulated sunlight for 60 min (100 mW cm$^{-2}$, AM 1.5). Following the light soak (yellow shaded area in Fig. 3f), we observe PL enhancements consistent with those reported in Fig. 2c, then we monitor the local retention by taking fluorescence images over extended time intervals (Fig. 3b–d; see Supplementary Fig. 10 for complete series, Supplementary Note 11), where the sample was kept in the dark in between imaging. In Fig. 3f we show the average PL relaxation dynamics of the dark, intermediate and bright regions of interest (ROIs) defined in Fig. 3e (Supplementary Fig. 10) as well as the average relaxation (dotted black trace) across the fluorescence image. The emission from the dark regions relax and eventually reach a stabilized emission after $\sim 9$ h, maintaining a PL level $\sim 3 \times$ higher than their initial value before illumination. In contrast, the emission from the bright regions relax and stabilize at a lower PL level (0.66) relative to their initial value. Although the emission from all regions generally improves while under illumination, these results suggest that the photo-induced cleaning is locally retained (when left in the dark) in dark regions but not in bright regions (Supplementary Figs 10 and 11; Supplementary Note 11). Importantly, the stabilized average PL intensity of the analysed area after $\sim 9$ h is approximately the same as the average initial PL intensity before illumination (initial and final points of dotted black trace). These observations are consistent with a new equilibrium corresponding to a redistribution of emission intensities with improved grain-to-grain homogeneity, which is only reached after an extended illumination period and several hours in the dark (Supplementary Fig. 12).

**Compositional changes after illumination**. To ascertain whether the light-induced PL enhancements and redistributions can be directly related to compositional changes, we now investigate the changes in local composition due to illumination by performing ToF-SIMS depth profiling on photo-irradiated films. We first light-soaked a local dark spot for several minutes with pulsed excitation (1.2 kJ cm$^{-2}$) and recorded the typical slow rises in emission (Fig. 4a). Previously, we have reported enhancements in solar cell performance metrics, namely the open-circuit voltage ($V_{oc}$), under similar photon doses used for this experiment[37]. We then performed ToF-SIMS on this same film and report the summed iodide signal through a depth profile at the illuminated region in Fig. 4b. Figure 4c shows the iodide distribution across the line scan (blue arrow) from Fig. 4b, along with the measured

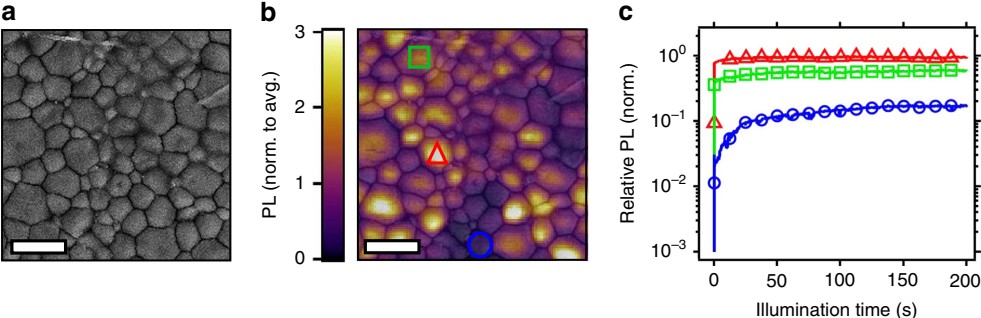

**Figure 2 | Local changes in photoluminescence (PL) over time under illumination.** (**a**) Correlated scanning electron microscopy (SEM) image and (**b**) fluorescence image of a perovskite film measured in nitrogen with pulsed photo-excitation (470 nm, 40 MHz repetition rate, 0.03 μJ cm$^{-2}$ per pulse) with semitransparent SEM image overlaid, scale bars, 2 μm. (**c**) PL intensity over time from a dark spot (blue circles, enhancement of 8.7 ×), intermediate spot (green squares, enhancement of 1.6 ×) and a bright spot (red triangles, enhancement of 1.4 ×) corresponding to the regions highlighted with the same symbols in **b**, with photo-excitation at 532 nm (188 mW cm$^{-2}$, ∼3 suns).

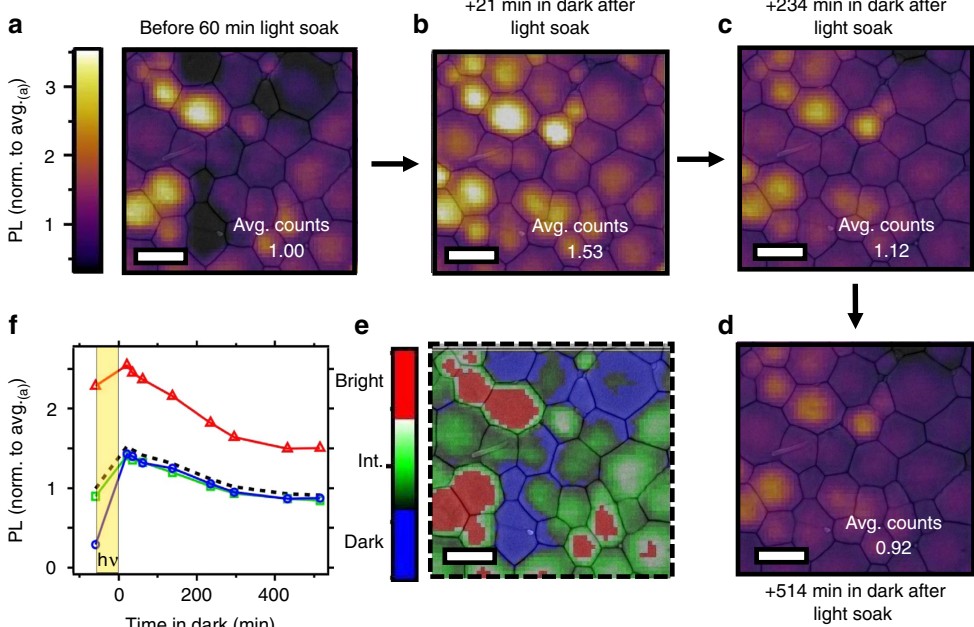

**Figure 3 | Local photoluminescence (PL) rises and relaxation after exposure to simulated sunlight.** Fluorescence images under pulsed excitation (470 nm, 40 MHz repetition rate, 0.03 μJ cm$^{-2}$ per pulse) measured in nitrogen with semitransparent scanning electron microscopy (SEM) images overlaid (**a**) before light soaking, and after exposing the entire film to simulated sunlight (AM 1.5, 100 mW cm$^{-2}$) for 60 min and leaving in the dark for (**b**) 21 (**c**) 234 and (**d**) 514 min (all images have the same PL intensity scale normalized to the average PL intensity in **a**, scale bars, 1 μm). (**e**) Three-colour scale image showing the regions classified as dark, intermediate (Int.) and bright (Supplementary Fig. 10). (**f**) Local PL enhancement and relaxation for dark (blue, enhancement of 4.9 ×), intermediate (green, enhancement of 1.6 ×) and bright (red, enhancement of 1.1 ×) regions, where the time (*t*) under illumination is highlighted by the yellow shaded region for −60≤*t*≤0 min, and *t*>0 show the local PL relaxation dynamics over time left in the dark. The dotted black line is the PL relaxation averaged across the whole fluorescence image.

laser profile (blue line). We see that the regions of highest illumination intensity corresponding to the spatial profile of the laser show depleted levels of iodide, while the immediately adjacent regions show an enrichment in iodide relative to the background iodide levels (ROI shown in Supplementary Fig. 13). These results suggest a lateral migration of iodine away from the illumination area, which can be attributed to iodine redistribution (Supplementary Fig. 13; Supplementary Note 12). The ToF-SIMS counts suggest that the relative changes in iodine-containing fragments are on the order of a few percent, and since we also observe local variations in other lower intensity fragments, we conclude that iodine migration alone does not encompass the complexity of this ionic redistribution effect (Supplementary Note 12).

To probe the extent of halide redistribution as a function of film depth, we illuminated two different regions, one with a photon dose of 1.2 kJ cm$^{-2}$ (red circle) and another with 2.4 kJ cm$^{-2}$ (green circle), as shown in Fig. 5a, and we show the iodide intensity map in Fig. 5b. We again observe the typical PL rises and an anti-correlation between the measured excitation laser beam spatial profile and the resulting iodide intensity (Supplementary Fig. 14). In Fig. 5c, we show ToF-SIMS depth profiling data for iodide. To highlight the differences relative to a control area, the data are displayed as the difference between the light-soaked regions (red and green lines) and the background iodide level from a region that had not been light-soaked (blue line). We also show the carrier generation profile through the film thickness as a guide to the eye. As seen in Fig. 5c, the

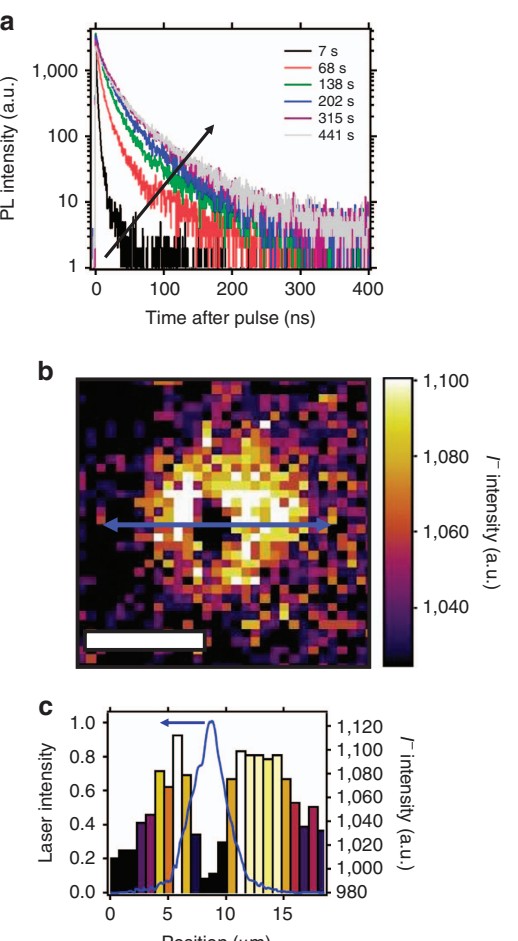

**Figure 4 | Iodide redistribution after light soaking.** (**a**) A series of time-resolved photoluminescence decays from a CH$_3$NH$_3$PbI$_3$ film measured over time under illumination before time-of-flight secondary-ion-mass spectrometry (ToF-SIMS) measurements. The sample was photoexcited with pulsed excitation (470 nm, 1.2 kJ cm$^{-2}$). (**b**) ToF-SIMS image of the iodide (I$^-$) distribution summed through the film depth (the image has been adjusted to show maximum contrast), scale bar, 10 μm. (**c**) Line scan of the blue arrow in **b** to show the iodide distribution (right axis). The measured spatial profile of the illumination laser (blue) is shown on the left axis.

iodide level is initially lower in the light-soaked areas in comparison with the background region, and this is particularly true for film depths corresponding to high carrier densities. Consistent with this observation, as the carrier generation profile decays through the film, the iodide level increases relative to the background level deeper in the film and at the substrate interface (Supplementary Fig. 14). We observe a greater fraction of iodide deeper in the film when exposed to a photon dose of 2.4 kJ cm$^{-2}$ (green line) as compared with 1.2 kJ cm$^{-2}$ (red line), suggesting that the magnitude of iodide displacement is dependent on photon dose. Figure 5d shows the depth profiles of regions adjacent to the illuminated areas (pink and gold lines), where the iodide content is higher in the bulk of the film as well as deeper in the film in comparison to the background profile. These data are consistent with iodine migration away from the illuminated region both laterally and vertically. We also observe similar effects in mixed halide (that is, CH$_3$NH$_3$PbI$_{3-x}$Br$_x$) films[19] (Supplementary Fig. 15) and also from energy-dispersive X-ray spectroscopy (EDS) measurements (Supplementary Fig. 16; Supplementary Note 13).

**Proposed mechanism of photobrightening.** Our interpretation of these results is that the illumination induces a redistribution of iodide resulting in a photo-cleaned (brightened) region with net reduced trap densities. The redistribution in iodide intensities alone is larger than the changes in iodine concentration associated with trap annihilation ($\sim$p.p.m.). Here we propose a mechanism in Fig. 6 that broadly describes our observations in this study, but it is likely that there are other complex mass transport mechanisms simultaneously occurring. On film formation, some regions contain larger densities of electronic traps, which could arise from iodine vacancies (undercoordinated lead sites, $\delta^+$) with associated interstitial iodide ions[8,18,29,38–40], which are numerous at surfaces and grain boundaries (Fig. 6a) (refs 10,41,42). Under illumination, we create a relatively high density of photo-excited electrons and holes, which is highest at the surface and falls off exponentially through the film (Fig. 5c). Many of the photo-excited electrons will become trapped in the vacancies particularly near the surface (Fig. 6b) (refs 13,29,42). The trap filling would perturb the system and create electric fields that induce iodide migration and subsequent trap annihilation in several ways: (1) Coulombic repulsion between now-unscreened iodide ions, (2) Spatial charge separation arising from the surface-trapped electrons and diffused holes or (3) A change in band bending at the surface on illumination[29]. In all cases, this field-induced migration allows the large amount of mobile iodide to fill vacancies and therefore yield a net reduction in the density of vacancies and interstitials which could be responsible for the non-radiative recombination (Fig. 6c)[39,40,43,44]. By locally illuminating a region we observe both vertical and lateral migration away from the carrier generation profile (cf. Fig. 5). If the entire film is illuminated (cf. Fig. 3), we expect vertical migration to still be prevalent and annihilation of defects under the illumination spot, leading to a net brightening of all grains, but the lateral migration will be dominated by local iodide and (filled) defect density gradients which will vary within and between grains; thus, some regions will locally brighten more than others due to relatively greater excess iodine removal in those regions. At higher excitation intensity, the time to reach stabilized emission will be faster because more traps will be filled by photo-generated electrons, thereby producing a larger field to drive halide migration. Likewise, at higher temperature the ionic conductivity will increase and the system will again reach stabilized emission in a shorter time. In the dark, the photo-excitation profile is removed and some traps remain, which may allow some iodide ions to slowly migrate back laterally or vertically to establish a new equilibrium (Fig. 6d; cf. Fig. 3), thereby leading to a partial reversibility. However, new concentration gradients may appear over time as ions cluster or disperse, thus the dynamics may be much more complicated (Supplementary Fig. 11).

We believe that our mechanism is also consistent with the rapid photo-brightening in perovskite nanocrystals and nano-crystalline domains with few trap sites per crystal, as reported by Tian *et al.*[43] and Tachikawa *et al*[31]. Here traps are efficiently filled and the gradient driving ion migration results in the rapid net removal of the few interstitial-vacancy pairs per crystal. The mechanism we propose here may not be able to explain all photo-brightening phenomena. For example, it has been recently shown that oxygen species could contribute to the photo-brightening process[14,30]. Tian *et al.*[14] recently proposed a model for the photo-brightening in the presence of oxygen, where photo-generated charge carriers and oxygen-related species interact to deactivate the trapping defects. In their mechanism and in ours, the photo-brightening is limited by the interaction volume of photo-excited carriers in the film. We take means to minimize oxygen in our measurements presented here, suggesting that oxygen may not be essential in the process. Nevertheless, the

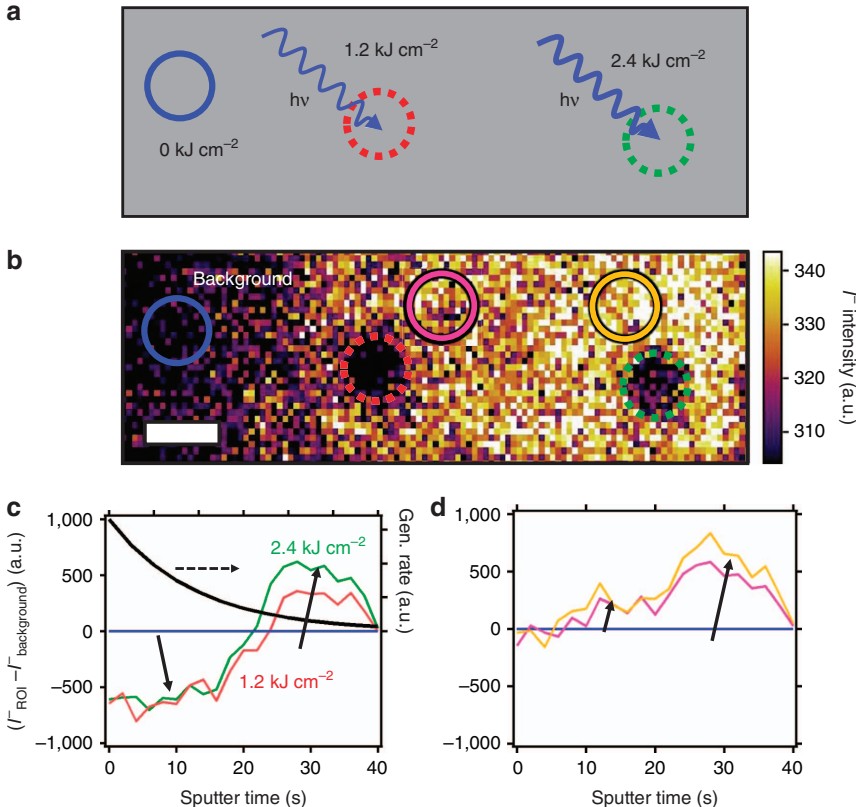

**Figure 5 | Time-of-flight secondary-ion-mass spectrometry (ToF-SIMS) depth profiling. (a)** Schematic of photon dose-dependent experiment indicating the regions exposed to $1.2 \, kJ \, cm^{-2}$ (red dotted circle) and $2.4 \, kJ \, cm^{-2}$ (green dotted circle) **(b)** ToF-SIMS image of the iodide ($I^-$) intensity distribution summed through the film depth (the image has been adjusted to show maximum contrast) after local exposure to the photon doses shown in **a**, scale bar, $5 \, \mu m$. **(c)** Depth profile data of the iodide intensity in regions of interest (ROI) relative to the background ($I^-_{ROI}$-$I^-_{background}$) for regions illuminated in **a**, with the carrier generation (gen.) rate plotted on the right axis. **(d)** Depth profile data for ROI's adjacent to $1.2 \, kJ \, cm^{-2}$ (pink) and $2.4 \, kJ \, cm^{-2}$ (gold) illumination areas compared with a background level (blue circle). All selected regions contained the same number of pixels to allow comparison.

competition between oxygen and iodine in both diffusion and ability to annihilate traps, and any possible synergy between the two, remains an ongoing question for the community.

## Discussion

Tian *et al.* and others also speculate that the photo-induced brightening could be related to ion migration[14,19]. Here we provide direct strong visual evidence of photo-induced halide redistribution effects. Hoke *et al.*[19] used PL measurements to calculate a bulk activation energy of $0.27 \pm 0.06 \, eV$ for halide migration in mixed halide systems ($CH_3NH_3PbI_{3-x}Br_x$), which roughly agrees with the value we extract here of $0.19 \pm 0.05 \, eV$ for the pure trihalide systems (Fig. 1c). Indeed, activation energies for iodide migration in $CH_3NH_3PbI_3$ have been recently estimated to be in the range $\sim 0.1$–$0.6 \, eV$ (refs 18,36). The differences may in part be due to different local stoichiometry, that is, iodide-rich or iodide-poor conditions, which has a large influence on the formation energies of iodide interstitials[39,45]. For example, the formation energy of iodide interstitials in iodide-rich conditions is also $\sim 0.2 \, eV$, suggesting that the activation energy we extract here could be related to the interstitial/vacancy-mediated transport of iodide[18,36]. The reversible ion redistribution could also be consistent with the photo-induced structural changes reported elsewhere[46,47].

Finally, we discuss how these slow transient effects and our measurements are relevant to full optoelectronic devices. We have observed a light-induced halide migration in neat films, which we attribute to the formation of local fields created by filling traps under illumination. Others have reported methylammonium

migration but only in full devices in the presence of a field across the entire film[24–26]. Since the migration in both cases likely arises from electric fields, we could use the light-induced changes as a proxy to learn about the field-induced changes in the full devices. For instance, the depth profile data (Fig. 5) revealed that just by illuminating with light, we can deplete iodine at the top surface and redistribute it to the substrate interface (that is, contact in a full device), which would strongly affect band bending and charge collection[16]. Furthermore, we learn that the ion migration in full devices is not simply due to defects moving under an applied field, rather there is a dynamic interplay between carriers, traps and mobile ions that needs to be considered. This has direct relevance to the hysteresis phenomena in full solar cells, which has been proposed to be primarily caused by ion migration[16,21,23] but also necessitates the presence of trap states[22]. The rise times to stabilized efficiency at the maximum power point in full n-i-p solar cells closely match the time scales to reach stabilized emission due to photo-induced cleaning under 1-sun equivalent irradiation[8,23], which is consistent with both phenomena arising from ion migration. Notably, we observe here that the halide migration is related to the passivation, or deactivation of traps, hence the hysteresis observed in solar cells could be amplified by the activation or deactivation of electron traps depending on whether the mobile iodine is pushed towards, or away from, the n-type charge collection layer. This is consistent with our recent work where we have reduced the perovskite trap density by an order of magnitude by treating the polycrystalline film surface with small molecules, such as pyridine, and we observe that the rise times to stabilized maximum power output are much faster

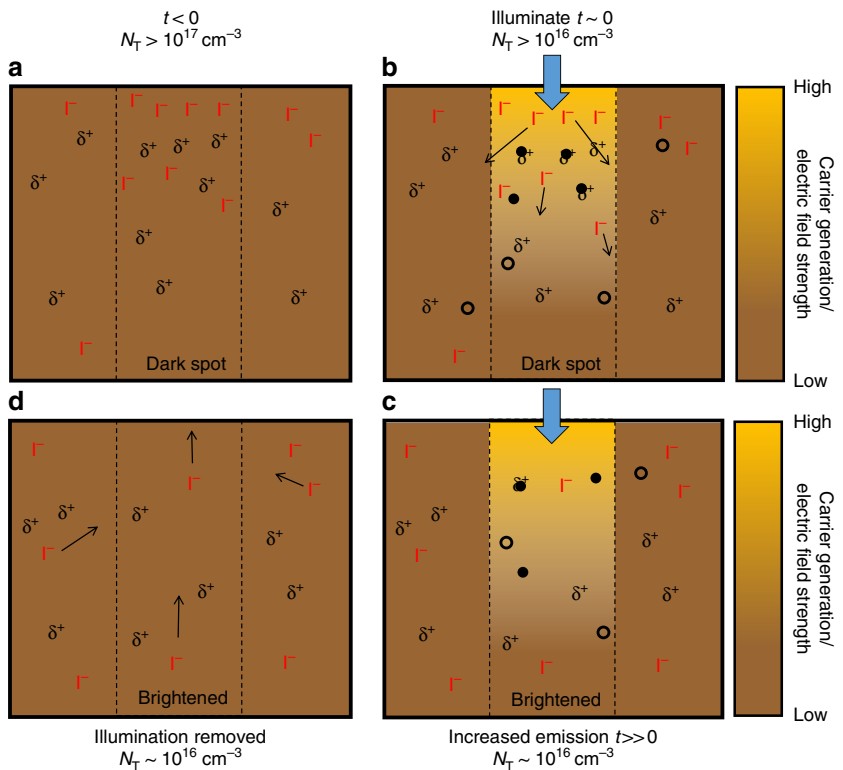

**Figure 6 | Proposed mechanism of photo-induced cleaning by halide redistribution.** The cartoon represents a cross-section of the films illuminated through the top surface. (**a**) The trap density in a 'dark spot' is initially high with a corresponding excess of iodide. (**b**) Upon illumination, electrons will quickly fill traps, inducing an electric field that causes iodide to migrate away from the illuminated region and fill vacancies. (**c**) The system eventually reaches a stabilized emission output with a reduced trap density and iodide concentration in the illuminated region. (**d**) When the illumination is removed, there may be concentration gradients driving some iodide back into the dark spot before it eventually reaches a new equilibrium with a net redistributed iodide profile. $\delta^+$ are iodide ($I^-$) vacancies, filled circles are electrons, open circles are holes.

and hysteresis is reduced[40,48]. In the extreme case of negligibly low trap densities as in single crystals, we show that the emission stabilizes rapidly (see Supplementary Fig. 1), and this is consistent with the minimal hysteresis effects reported for devices comprising single crystals or millimetre-scale grain thin films[32,33,49]. We can conclude that the rise times to stabilized maximum power are much faster for solar cells incorporating perovskites with lower trap densities, and a lack of transient behaviour strongly suggests that ion migration has been suppressed. It is therefore imperative to reduce the trap density to eliminate unwanted ion migration effects, and future work should concentrate on fabricating perovskites with uniform iodine distributions and low trap densities (by improving growth processes or post-treating samples) to achieve instantaneously stabilized optoelectronic behaviour. It still remains unclear if we require films with a stoichiometric composition, or if in fact halide-deficient regions are more PL active[39]. The finding that potentially large fractions of iodine are moving under illumination without major changes to crystal structure (cf. X-ray diffraction measurements, Supplementary Fig. 3) needs to be understood and remains an open question for the community.

In conclusion, we have investigated $CH_3NH_3PbI_3$ films without contacts and find that the PL lifetime and PL intensity rise substantially over time under illumination corresponding to nearly an order-of-magnitude reduction in trap densities. On the microscale, we find that the changes in PL and trap density are particularly significant for the dimmer spots. There is a strong correlation between the increase in PL over time under illumination and a redistribution of iodine away from the illuminated region, where this redistribution is likely connected to the redistribution of stabilized PL intensities. These results are consistent with a photo-induced iodide migration with an activation energy of $0.19 \pm 0.05$ eV. This work gives direct visualization of photo-induced halide migration in the neat perovskite materials that is related to slow transient optoelectronic phenomena and lowering of the trap densities through 'cleaning' effects; the precise mechanism of migration and significance of the activation energy remains an open question for the community. The correlations between local photophysical behaviour and local composition profiled through the film presented in this work will provide an excellent tool set to further understand the impact of chemical-, photo-, field- and atmospheric-induced effects on these slow transients and ion migration in operating devices.

## Methods

**Sample fabrication.** Perovskites were prepared using a method described elsewhere where a methylammonium iodide ($CH_3NH_3I$) and lead acetate $Pb(Ac)_2 \cdot 3H_2O$ precursor mixture was employed[29]. To generate the perovskite solution, $CH_3NH_3I$ (Dyesol) and $Pb(Ac)_2 \cdot 3H_2O$ were dissolved in anhydrous $N,N$-dimethylformamide at a 3:1 molar ratio with final concentration of $\sim 30$ wt%, and the stabilizer hypophosphorous acid (HPA) was added at a molar ratio of 7.5% with respect to $Pb(Ac)_2 \cdot 3H_2O$. $Pb(Ac)_2 \cdot 3H_2O$ (316512) and HPA (214906) were purchased from Sigma Aldrich. Microscope slides and coverslips were washed sequentially with soap (2% vol. Hellmanex in water), de-ionized water, isopropanol, acetone and finally treated under oxygen plasma for 10 min. The precursor solution was spin-coated at 2,000 r.p.m. for 45 s in a nitrogen-filled glovebox, and the substrates were then dried at room temperature for 10 min

before annealing at 100 °C for 5 min. The samples were then stored in a nitrogen-filled glovebox until used.

**Photoluminescence measurements.** Bulk time-resolved PL (TRPL) decays were acquired using a time-correlated single photon counting (TCSPC) setup (FluoTime 300, PicoQuant GmbH). Temperature-dependent measurements were carried out *in vacuo* using an Oxford Instruments OptistatDN cryostat with a specialized fitting for the TCSPC set-up. Samples were photoexcited using a 507-nm laser head (LDH-P-C-510, PicoQuant GmbH) with pulse duration of 117 ps, fluences of ~0.03–3 μJ cm$^{-2}$ per pulse, and a repetition rate of 1 MHz. To acquire PL decays over time under illumination, integration times were kept short (2–10 s). The stated times in the legends are time stamps at the end of each integration window for each curve.

**Confocal fluorescence imaging.** Optical microscopy and spectroscopy were performed under nitrogen (flow cell) using a custom sample scanning confocal microscope with TCSPC (PicoHarp 300) capabilities built around a Nikon TE-2000 inverted microscope fitted with an infinity corrected ×50 dry objective (Nikon L Plan, NA 0.7, CC 0–1.2). A 470-nm pulsed diode laser (PDL-800 LDH-P-C-470B, 350 ps pulse width) was used for excitation with a repetition rate of 1 MHz for time-resolved PL measurements and 40 MHz for collecting fluorescence images. A 532-nm CW laser (CrystaLaser, GCL532-005-L) was used to excite local regions and monitor the PL rises over time. Samples were excited face-on through the flow cell's coverslip and the emission was filtered through a 700-nm longpass filter then directed to a Micro Photon Devices (MPD) PDM Series single photon avalanche photodiode with a 50-μm active area. The sample stage was controlled using a piezo controller (Physik Instrumente E-710) with custom software. For collecting fluorescence images, the pixel size was 100 nm with a short pixel dwell time (integration time) of 50 ms, to minimize the extent of photoinduced cleaning prior to collecting longer exposure local measurements. For wide area light-soaking experiments, the entire film was exposed to AM 1.5 G light illumination from a solar simulator (Solar Light Co. model 16S-300) with intensity calibrated to 100 mW cm$^{-2}$ using a Si reference cell and source meter (Keithley, 2400 Series). Before collecting fluorescence images, the system was calibrated using 200 nm fluorescent microspheres (Life Technologies FluoSpheres Polystyrene Microspheres, 200 nm, red fluorescent, 580/605) to yield a point-spread function of ~350 nm at full width at half maximum. To correlate fluorescence images with SEM micrographs and ToF-SIMS, we made fiducial markers on the sample and used these markers along with local microstructure to match areas across the separate measurements.

**Scanning electron microscopy.** SEM images were taken using a FEI Sirion SEM at 5–10 kV accelerating voltage. To prevent charging effects, samples for SEM were imaged after sputtering ~7 nm of Au/Pd using a SPI-Module Sputter Coater with argon flow.

**Time-of-flight secondary-ion-mass spectrometry.** ToF-SIMS depth profiles were carried out using an IonToF ToF-SIMS 5 instrument. Dual-beam depth profiles were acquired using a 25-keV Bi$_3^+$ ion in delayed extraction mode for images and a 20-keV argon 1000 gas cluster ion source for sputtering. Sputtering was carried out over a 600 micron × 600 micron area at 3.3 nA current for 2 s for a dose of 1.1 × 10$^{13}$ ions cm$^{-2}$ per sputter cycle. Images were acquired over a 50 micron × 50 micron area within the centre of the sputter crater with a current of ~0.03 pA for a dose per cycle of 5.7 × 10$^{11}$ ions cm$^{-2}$. Secondary ions were detected using a time-of-flight mass analyzer. Due to non-linearity in the mass axis introduced when using delayed extraction[50], peak identities were determined from spectra acquired using the standard spectroscopy mode of the IonToF instrument using similar settings as described above. The negative ion spectra were calibrated using the CH$^-$, OH$^-$ and I$^-$ peaks. Calibration errors were kept below 30 p.p.m. Mass resolution ($m/\Delta m$) for a typical spectrum was between 4,000 and 6,000 for $m/z$ 25.

**Data availability.** The data that support the findings of this study are available from the corresponding author upon request.

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

## Acknowledgements

The research leading to these results has received funding from the European Union Seventh Framework Programme (FP7/2007–2013) under grant agreement 604032 of the MESO project, and also from the People Programme (Marie Curie Actions) of the European Union's Seventh Framework Programme (FP7/2007–2013) under REA grant agreement number PIOF-GA-2013-622630. D.W.D. and D.S.G. acknowledge DOE (DE-SC0013957) for supporting the local imaging work. D.W.D. acknowledges support from an NSF Graduate Research Fellowship (DGE-1256082) and thanks M. Ziffer and J. Mohammed for providing single crystal samples. S.D.S. also thanks R.H. Friend for additional support and R. Brenes for additional PL measurements. W.Z. thanks the EPSRC Supergen Supersolar project for financial support. This research was performed in part at the Molecular Analysis Facility (MAF) which is funded, in part, by the University of Washington, Molecular Engineering and Sciences Institute, and the Clean Energy Institute, as well as infrastructure grants from the National Institutes of Health (NIH) and the National Science Foundation. The argon cluster source used in this research was funded by NIH grant S10 OD010607. V.B. was supported as part of the Center for Excitonics, an Energy Frontier Research Center funded by the U.S. Department of Energy, Office of Science, Office of Basic Energy Sciences under Award Number DE-SC0001088 (MIT). The authors thank MIT Libraries for contributions to open access article processing fees.

## Author contributions

The project was conceived, planned, and co-ordinated by S.D.S., D.W.D., H.J.S. and D.S.G. Samples were prepared by W.Z., bulk PL measurements were performed by S.D.S. and D.W.D., micro-scale PL by D.W.D. and ToF-SIMS measurements by D.W.D. and D.J.G. V.M.B. developed the mathematical formulation of the trap model. All authors assisted in the interpretation of the results. S.D.S. and D.W.D. wrote the manuscript and all authors helped in editing.

## Additional information

**Competing financial interests:** The authors declare no competing financial interests.

**How to cite this article:** deQuilettes, D. W. *et al.* Photo-induced halide redistribution in organic–inorganic perovskite films. *Nat. Commun.* 7:11683 doi: 10.1038/ncomms11683 (2016).

