## [Peer Review File · Nature Communications]

Reviewer #1 (Remarks to the Author):

The changes the authors have done in the revised version of the manuscript by seQuillettes et al did not fully address the concern expressed in my report on the original paper. The paper still contains statements which are not supported by the experimental data and the main experimental result (TOF-SIMS measurements) is incorrectly connected to the photo-brightening. Although the experimental results are potentially interesting, their publication in Nat Comm only possible if an adequate explanation is presented and the results are adequately placed in the framework of the existing literature data.

1) The main issue concerns the interpretation of TOF-SIMS results.

The authors agree that "It is not possible to tell if the detected I-, I₂- came from a defect I- or it is just an iodine broken out by the ion beam from the perfect perovskite unit cell."

They also argued that the accuracy is very good and can detect 1 ppm of iodide content change. Good, I can believe this. However, it still does not answer my question of the data interpretation. I was simply not able to formulate it clearly last time.

As it is known and also discussed in the paper, the concentration of the traps is very low. Basically one trap per many-many of unit cells. Concentration of iodine in the pure perovskite is about $1.25 \times 10^{22} \text{ cm}^{-3}$. The initial concentration of traps is (see page 5) is about 10^{17} cm^{-3} . So, one trap per 100 000 iodine atoms.

Yes, it is possible to detect 1 per 100 000 as pointed out by the authors.

However the difference between illuminated and un-illuminated areas has to be less than the concentration of the defects. (because it is only the defects which move according to the authors' model).

The signal in TOF-SIMS is proportional to the concentration of the particular atom (iodide in our case). Let us look at Fig. 4. It shows the signal from iodide. The difference between the illuminated and non-illuminated areas should be less than the total concentration of the traps (1 per 100 000 atoms).

So, we expect

$\text{Signal (I, non-illuminated)} - \text{Signal (I, illuminated)} / \text{Signal (I, non-illuminated)} < 0.00001$

However, the data from the figure gives the same parameter of about $100/1000 = 0.1$

So, whatever the authors detected, it is not related to the defect sites migration which concentration they assumed to be $10\text{-}15 \text{ cm}^{-3}$ or so. The authors observed really large changes in the structure and chemistry of the material, but is it definitely not the migration of that tiny amount of iodide. The changes might be related only to the surface layer which makes the probability of ion extraction different in the illuminated and non illuminated area. The sensitivity of the method to the surface has not been discussed in the manuscript.

Anyway, without explanation of the absolute numbers (or the relative change of the signal as I suggested) the paper cannot be accepted.

Some other issues.

2) About the novelty.

The authors: "There has not yet been decisive evidence presented or model proposed that specifically isolates whether optoelectronic improvements are a result of an increase in the

radiative bimolecular rate or a reduced non-radiative rate constant. Using our kinetic model, we unequivocally correlate the improvement to the latter (i.e. a reduction in trap state density) and for the first time quantify these changes. We have changed the sentence in the abstract to make these points clear:

"We demonstrate that the photo-induced "brightening" of the perovskite PL can be attributed to an order-of-magnitude reduction in trap state density."

Well, I am not sure I see the novelty here. There is absolutely no way to suggest that the observed photo-brightening is due to an increase in the radiative bimolecular rate. This is simply because we know (ref. 14 for example) that the PL lifetime increases upon light irradiation (instead of decreasing if the radiative rate would increase). Increasing of PL lifetime can be only due to decreasing of the concentration of traps.

So, the authors seem to assign some novelty to a common text-book knowledge and single-step logic.

This trivial conclusion has been already been discussed in ref. 14 by Tian et al. Moreover, photobrightening in semiconductor QD and other structures was always related to trap passivation by some surface photochemistry. This again means reducing of the effective trap concentration.

In the reply letter the authors wrote:

"We agree that changed behavior under illumination has been generally reported but, to the best of our knowledge, our study is the first to correlate these different rise times to changes in composition and local trap state densities. "

This is not correct. The difference in the rise times and in general the connection of the effect of photo brightening to change of trap concentration has been already discussed in the literature.

I looked carefully at the ref. 14, Tian et al. It is proposed there that light induces modification of the trapping sites making them inactive. There is even a model which is based on a reaction zone of un-active defects which spreads in the crystal.

3) The principle difference between the model suggested in ref. 14 and the one presented in the manuscript is that in the former the concentration of the active traps is assumed to reduce in the whole sample, while the authors here try to just re-distribute the traps over the crystal.

In the recent paper by the same authors as ref. 14, Tian et al, Enhanced Organo-Metal Halide Perovskite Photoluminescence from Nanosized Defect-Free Crystallites and Emitting Sites, *J. Phys. Chem. Lett.*, 2015, 6 (20), pp 4171-4177

photo-brightening of very small crystals is discussed. In crystals of 100 nm in size where there is no place for the iodide to migrate. However, very fast brightening in such small crystals was reported. So, the migration model presented by the authors does not work even hypothetically here.

4) In response to my question regarding this issue the authors added the following paragraph to the text:

"By locally illuminating a region we observe both vertical and lateral migration away from the carrier generation profile (cf. Figure 5). If the entire film is illuminated (cf. Figure 3), we expect vertical migration to still be prevalent, leading to a net brightening of all grains, but the lateral migration will be dominated by local iodide and (filled) defect density gradients which will vary within and between grains; thus, some regions will locally brighten more than others due to

relatively greater excess iodine removal in those regions."

As we all know, the charge migration length in perovskite is usually trap-limited. If the concentration of the traps is low then charges migrate over 1 micrometer or more. So, if we do not really de-activate the PL quenching defects but just move them to a different place, they always can be reached by charge carriers. So, moving the defects around in space without modification them cannot give a strong photobrightening effect.

5) To conclude, the paper contains interesting experimental observation, however, their interpretation is contradictory and unsatisfactory. I suggest reconsidering this paper in Nature Comm. if the authors:

1. Totally reconsider the interpretation of the TOF results.
2. Reconsider the model, make it possible to explain photo-brightening of nano-sized crystals (Tain et al JPCLetters) and discuss the literature data where some models have been already presented (e.g. ref 14). Regardless with oxygen or without. Nitrogen-filled box used by the authors to prepare the sample does not totally exclude oxygen from the material. This is simply impossible.
3. Most probably it will not be possible for the authors to explain all observations in one model. However, admitting that the current status of the knowledge does not allow suggesting the definite mechanism of the photo-brightening in perovskites does not make the paper less interesting for the community.

Reviewer #2 (Remarks to the Author):

I have read the manuscript in its revised form and the response to reviewers submitted by the authors. The authors have convincingly answered to the major points raised by the reviewers and the manuscript is definitely improved, also including additional data on single crystals that confirm the original conclusions about light-induced defect migration.

I believe the manuscript should now be published in Nature Comms. without further revisions.

Extra remarks from reviewer #2

The reviewer argument could be reasonable, and it requires additional explanations from the authors.

If the TOF-SIMS signal is linearly dependent on total iodine concentration locally probed when scanning the position (Figure 4c), what the authors are measuring is a much larger variation than that expected based on the variation in trap density upon illumination. This latter quantity is correctly estimated by the reviewer to be less than $10^{17} / 10^{22} = 10^{-5}$, while the signal variation from Figure 4 is 1/10.

I am not saying the authors have misinterpreted the data here (there could be a volume factor that has not been included or explained or the signal may not be linearly dependent on concentration), but a further explanation is definitely in order. To make things clearer, I suggest to report data in Figure 4 (and 5) on a relative scale (i.e. % iodine enrichment) which is directly related to concentration, rather than the signal itself.

"The signal in TOF-SIMS is proportional to the concentration of the particular atom (iodide in our case). Let us look at Fig. 4. It shows the signal from iodide. The difference between the illuminated and non-illuminated areas should be less than the total concentration of the traps (1 per 100 000

atoms).

So, we expect

Signal (I, non-illuminated) - Signal (I, illuminated) / Signal (I, non-illuminated) < 0.00001

However, the data from the figure gives the same parameter of about $100/1000 = 0.1$

So, whatever the authors detected, it is not related to the defect sites migration which concentration they assumed to be $10\text{-}15\text{ cm}^{-3}$ or so. The authors observed really large changes in the structure and chemistry of the material, but is it definitely not the migration of that tiny amount of iodide. The changes might be related only to the surface layer which makes the probability of ion extraction different in the illuminated and non illuminated area. The sensitivity of the method to the surface has not been discussed in the manuscript.

Anyway, without explanation of the absolute numbers (or the relative change of the signal as I suggested) the paper cannot be accepted.

Reviewer #3 (Remarks to the Author):

The manuscript was revised accordingly and is now suitable for publication without further revision.

Reviewers' comments:

Reviewer #1 (Remarks to the Author):

The changes the authors have done in the revised version of the manuscript by seQuillettes et al did not fully address the concern expressed in my report on the original paper. The paper still contains statements which are not supported by the experimental data and the main experimental result (TOF-SIMS measurements) is incorrectly connected to the photo-brightening. Although the experimental results are potentially interesting, their publication in Nat Comm only possible if an adequate explanation is presented and the results are adequately placed in the framework of the existing literature data.

1) The main issue concerns the interpretation of TOF-SIMS results.

The authors agree that "It is not possible to tell if the detected I⁻, I²⁻ came from a defect I⁻ or it it just an iodine broken out by the ion beam from the perfect perovskite unit cell."

They also argued that the accuracy is very good and can detect 1 ppm of iodide content change. Good, I can believe this. However, it still does not answer my question of the data interpretation. I was simply not able to formulate it clearly last time.

As it is known and also discussed in the paper, the concentration of the traps is very low. Basically one trap per many-many of unit cells. Concentration of iodine in the pure perovskite is about $1.25 \cdot 10^{22}$ cm⁻³. The initial concentration of traps is (see page 5) is about 10^{17} cm⁻³. So, one trap per 100 000 iodine atoms.

Yes, it is possible to detect 1 per 100 000 as pointed out by the authors.

However the difference between illuminated and un-illuminated areas has to be less than the concentration of the defects. (because it is only the defects which move according to the authors' model).

The signal in TOF-SIMS is proportional to the concentration of the particular atom (iodide in our case). Let us look at Fig. 4. It shows the signal from iodide. The difference between the illuminated and non-illuminated areas should be less than the total concentration of the traps (1 per 100 000 atoms).

So, we expect

$\text{Signal (I, non-illuminated)} - \text{Signal (I, illuminated)} / \text{Signal (I, non-illuminated)} < 0.00001$

However, the data from the figure gives the same parameter of about $100/1000 = 0.1$

So, whatever the authors detected, it is not related to the defect sites migration which concentration they assumed to be 10^{15} cm⁻³ or so. The authors observed really large changes in the structure and chemistry of the material, but is it definitely not the migration of that tiny amount of iodide. The changes might be related only to the surface layer which makes the probability of ion extraction different in the illuminated and non illuminated area. The sensitivity of the method to the surface has not been discussed in the manuscript.

Anyway, without explanation of the absolute numbers (or the relative change of the signal as I

suggested) the paper cannot be accepted.

We first want to thank the referee for taking the time to clarify their point. This precise topic has been the subject of many of our internal discussions and the interpretation is unfortunately not trivial – though we agree with the referee that this needs to be better addressed in the manuscript.

We agree with the reviewer’s back-of-the-envelope approximation and that there is a discrepancy between the defect densities and the differences measured in iodide intensity. We do acknowledge that there may be a lot more iodide moving under illumination than just the iodide associated with the traps (~ppm). Since ToF-SIMS is not directly quantitative, we cannot calculate the absolute changes in iodine, but we have now approximated the relative change in intensities and explicitly discussed this in the text, added many of these details to the SI and added the following paragraph:

“The ToF-SIMS counts suggest that the relative changes in iodine-containing fragments are on the order of a few percent (see SI for details),....”

In the SI we have added:

“We identified and analyzed several different negative ion fragments and report the ToF-SIMS images most representative of the data set (Supplementary Fig. 13), including I^- , I_2^- , PbI_2^- , PbI_3^- , $Pb_2I_5^-$, $Pb_3I_7^-$, PO_3^- , and $C_3HN_2^-$ (a fragment of methylammonium). In Supplementary Fig. 13a, we show the integrated iodide counts summed through the film depth and also define regions of interest for the illuminated region (red circle), an adjacent region (green circle) and a background region far from illumination (blue circle). As ToF-SIMS is primarily a qualitative technique, obtaining absolute changes in iodine content is not possible without careful calibration, and is therefore beyond the scope of this work. In order to extract an approximate relative change in intensity of iodine-containing fragments (R) between light-soaked, adjacent, and background regions, we used the following equation:

$$R = \sum_i^N \left(\frac{M_i^-(ROI) - M_i^-(Backgrd)}{M_i^-(Backgrd)} \right) \times 100\%$$

where $M_i^- (...)$ denotes the intensity of the iodine-containing fragment in the region of interest. To a first-order approximation, we estimate $R = -1.2\%$ in the illuminated region and $R = +1.4\%$ in the adjacent region, indicating that iodine has been partially redistributed. These values are significant given we only observe ~0.5% variations in several background regions, though we emphasize again here that there is likely a large error without proper calibration.”

And also a few other clarifying sentences:

“The redistribution in iodide intensities alone is larger than the changes in iodine concentration associated with trap annihilation (~ppm).”

We have also performed an X-Ray Diffraction (XRD) study to further investigate any structural changes before and after illumination at a similar dose as applied to the samples for the SIMS measurements. We find no significant differences in the diffractograms within the variation and error of the measurement (Supplementary Fig. 3, Fig. R1 below). This suggests that the light-soaking does not induce any significant structural changes observable by this technique, and this is also consistent with the PL spectra not changing spectral shape or position over time under illumination (Supplementary Fig. 2).

Supplementary Figure R1. Monitoring structural changes of films under illumination. X-Ray Diffractograms (4 minute integration time) of a $\text{CH}_3\text{NH}_3\text{PbI}_3$ film acquired before illumination and during various windows while under constant illumination with a laser at a wavelength of 532 nm, with an intensity of (a) $\sim 600 \text{ mW/cm}^2$ (~ 10 sun equivalent, total dose of $\sim 0.7 \text{ kJ/cm}^2$) and $\sim 6000 \text{ mW/cm}^2$ (~ 100 sun equivalent, total dose of $\sim 3 \text{ kJ/cm}^2$). To within spot-to-spot variation of the measurement, no significant changes were observed.

We have also added the following line:

“We also do not observe any significant changes in X-Ray Diffraction (XRD) patterns of the films acquired while light soaking, indicating that the illumination is not inducing large structural changes (Supplementary Fig. 3)¹⁹.”

The changes might be related only to the surface layer which makes the probability of ion extraction different in the illuminated and non illuminated area. The sensitivity of the method to the surface has not been discussed in the manuscript.

We agree that these are good points and we did not explicitly discuss them in the previous version of the manuscript. ToF-SIMS is a surface sensitive technique probing the first few nanometers. We are

depth profiling through the sample by sputtering and at each depth the SIMS signal intensity is proportional to the composition at the top surface of the film after each sputter cycle. Therefore, we are able to get a representative picture of the signal intensity of the fragments through the film depth and not just the surface of the film. We have added remarks to the SI on the surface sensitivity of the technique as suggested:

"ToF-SIMS is a surface-sensitive technique with typical molecular ion escape depths of a few nanometers²⁷. For depth profiling, the signal intensity is proportional to the composition at the top surface of the film after each sputter cycle."

Regarding the suggestion of an alternate explanation for the local changes in the iodide intensity map based on differences in the probability of ion extraction in illuminated and non-illuminated areas. We do not believe we are observing an artifact from this possible scenario, as we would expect the adjacent region (red circle in Supplementary Fig. 13a), which has not been illuminated, to have a similar depth profile and intensity to the background region. In contrast, we still observe distinct profiles and intensities for the adjacent and background regions, which were both not illuminated. This strongly suggests that material is moving laterally outside the illumination region. In addition, there are no other high yield iodine-containing fragments with similar intensity maps as iodide; we would expect a systematic artifact in the other fragment intensity maps if illumination was changing the probability of ion extraction. We have included text addressing this concern in the SI:

"We briefly consider whether illumination could induce local variations in sputtering rates and ion extraction leading to artifacts in the intensity maps. In this possible scenario, we would expect the adjacent region (red circle in Supplementary Fig. 13a) to have a similar depth profile and intensity as the background region (blue circle in Supplementary Fig. 13a) – both of which have not been illuminated. In contrast, we still observe distinct profiles and intensities in the adjacent and background regions (Supplementary Fig. 14). This strongly suggests that material is moving laterally outside the illumination region. In addition, there are no other high yield iodine-containing fragments with similar intensity maps as iodide. If illumination was changing the sputtering rate and probability of ion extraction, we would expect a systematic artifact in at least some of the other fragment intensity maps."

As discussed further below, we have also changed the text throughout to better reflect the data presented, have clarified our interpretation, and also have made disclaimers before proposing our mechanism, which we believe to be the only part which connects iodide motion to trap populations.

Some other issues.

2) About the novelty.

The authors: "There has not yet been decisive evidence presented or model proposed that specifically isolates whether optoelectronic improvements are a result of an increase in the radiative bimolecular rate or a reduced non-radiative rate constant. Using our kinetic model, we unequivocally correlate the improvement to the latter (i.e. a reduction in trap state density) and for the first time quantify these changes. We have changed the sentence in the abstract to make these points clear:

"We demonstrate that the photo-induced "brightening" of the perovskite PL can be attributed to an order-of-magnitude reduction in trap state density."

Well, I am not sure I see the novelty here. There is absolutely no way to suggest that the observed photo-brightening is due to an increase in the radiative bimolecular rate. This is simply because we know (ref. 14 for example) that the PL lifetime increases upon light irradiation (instead of decreasing if the radiative rate would increase). Increasing of PL lifetime can be only due to decreasing of the concentration of traps.

So, the authors seem to assign some novelty to a common text-book knowledge and single-step logic.

We agree with the statements the referee has made.

We believe the novelty in this part of the manuscript is demonstrated in being the first to quantify the reduction in the trap density. In particular, we calculate an order of magnitude reduction in the trap state density, where the stabilized values are still well above the trap densities reported for single crystals. This information is important because it suggests that, although light-soaking leads to improvements in optoelectronic quality, the polycrystalline films still have significant scope for improvement by closing the gap in trap density with their single crystal counterparts. We have added additional text to highlight the implications of this result:

"Importantly, the stabilized values ($\sim 10^{16} \text{cm}^{-3}$) are still well above the trap densities reported for single crystals ($\sim 10^{10} \text{cm}^{-3}$),^{32,33} suggesting that the polycrystalline films still have significant scope for improvement and that light soaking alone may not close the gap in trap density with their single crystal counterparts."

This trivial conclusion has been already been discussed in ref. 14 by Tian et al. Moreover, photobrightening in semiconductor QD and other structures was always related to trap passivation by some surface photochemistry. This again means reducing of the effective trap concentration.

In the reply letter the authors wrote:

"We agree that changed behavior under illumination has been generally reported but, to the best of our knowledge, our study is the first to correlate these different rise times to changes in composition and local trap state densities. "

This is not correct. The difference in the rise times and in general the connection of the effect of photo brightening to change of trap concentration has been already discussed in the literature.

We agree that photobrightening has been *qualitatively* connected with trap concentration in the literature. We believe that our study is the first to:

1. Quantify the change in trap concentration (i.e. an order of magnitude reduction)
2. Systematically undertake the photobrightening measurements at different temperatures and different excitation intensities
3. Correlate the photobrightening with directly-observable changes in **chemical composition**.

I looked carefully at the ref. 14, Tian et al. It is proposed there that light induces modification of the trapping sites making them inactive. There is even a model which is based on a reaction zone of un-active defects which spreads in the crystal.

As the referee suggested at the beginning of their comments, we have now attempted to better frame our results in context with the existing literature data. We have added additional text to the manuscript describing the work by Tian et al which we believe, to the best of our knowledge, is the only other detailed mechanism in the literature for photobrightening of perovskites:

“The mechanism we propose here may not be able to explain all photo-brightening phenomena. For example, it has been recently shown that oxygen species could contribute to the photo-brightening process^{14,30}. Tian et al. recently proposed a model for the photo-brightening in the presence of oxygen, where photo-generated charge carriers and oxygen-related species interact to deactivate the trapping defects¹⁴. In their mechanism and in ours, the photo-brightening is limited by the interaction volume of photo-excited carriers in the film. We take means to minimize oxygen in our measurements presented here, suggesting that oxygen may not be essential in the process. Nevertheless, the competition between oxygen and iodine in both diffusion and ability to annihilate traps, and any possible synergy between the two, remains an ongoing question for the community.”

We believe that this makes the manuscript more representative and reiterates how our understanding of these complex photophysical and chemical phenomena as a community is moving forward.

3) The principle difference between the model suggested in ref. 14 and the one presented in the manuscript is that in the former the concentration of the active traps is assumed to reduce in the whole sample, while the authors here try to just re-distribute the traps over the crystal.

We apologise that our mechanism was unclear. We are in fact also proposing that the illumination leads to a net reduction of traps in the illuminated region of the sample rather than the traps just moving to different regions of the film. We propose that there are regions with large concentrations of iodide, for example potentially on the surfaces, with large trap densities associated with this excess iodide. The redistribution annihilates many of these traps, leading to a net reduction in trap density by an order of magnitude – as ascertained from bulk PL measurements and using our trap kinetic model. This would still be the case when the entire film is illuminated -- see for example Figure 3 where we have a net brightening across the film.

We now have a sentence that says,

“In all cases, this field-induced migration allows the large amount of mobile iodide to fill vacancies and therefore yield a net reduction in the density of vacancies and interstitials which could be responsible for the non-radiative recombination (Figure 6c)^{39,43}.”

We also demonstrate this graphically by showing trap annihilation in Fig 6b and a reduced number of vacancies and interstitials in 6c and 6d.

In fact, we believe that the model we propose here and that proposed by Tian et al. are self-consistent and compatible. As per the response to point 2, we have more explicitly referred to the model of Tian et al and how it relates to our model. It remains an interesting question to the community as to whether there is a competitive or even synergetic interplay between the iodide- and oxygen-induced photo-enhancements.

In the recent paper by the same authors as ref. 14,
Tian et al, Enhanced Organo-Metal Halide Perovskite Photoluminescence from Nanosized Defect-Free

Crystallites and Emitting Sites, J. Phys. Chem. Lett., 2015, 6 (20), pp 4171-4177

photo-brightening of very small crystals is discussed. In crystals of 100 nm in size where there is no place for the iodide to migrate. However, very fast brightening of such small crystals was reported. So, the migration model presented by the authors does not work even hypothetically here.

As mentioned above, our model describes the net removal of defects within the film, and we believe it applies to small crystals as well. In particular, Tian et al calculated only a few quenching sites per crystal. In our model, these traps would be rapidly passivated by the photoinduced migration of excess surface iodide into the bulk and subsequent neutralization of interstitial-vacancy pairs.

We have added a paragraph to this effect, where ref. 44 is the reference suggested by the referee above:

"We believe that our mechanism is also consistent with the rapid photo-brightening in perovskite nanocrystals and nanocrystalline domains with few trap sites per crystal, as reported by Tian et al.⁴⁴ and Tachikawa et al³¹. Here, traps are efficiently filled and the gradient driving ion migration results in the rapid net removal of the few interstitial-vacancy pairs per crystal."

4) In response to my question regarding this issue the authors added the following paragraph to the text:

"By locally illuminating a region we observe both vertical and lateral migration away from the carrier generation profile (cf. Figure 5). If the entire film is illuminated (cf. Figure 3), we expect vertical migration to still be prevalent, leading to a net brightening of all grains, but the lateral migration will be dominated by local iodide and (filled) defect density gradients which will vary within and between grains; thus, some regions will locally brighten more than others due to relatively greater excess iodine removal in those regions."

As we all know, the charge migration length in perovskite is usually trap-limited. If the concentration of the traps is low then charges migrate over 1 micrometer or more. So, if we do not really de-activate the PL quenching defects but just move them to a different place, they always can be reached by charge carriers. So, moving the defects around in space without modification them cannot give a strong photobrightening effect.

As our mechanism describes the removal of defects and not just the movement to new locations (see response to point 3), we believe we would still see a photobrightening effect from illuminating the whole sample.

We have slightly modified the text the referee mentioned to make this more clear.

"If the entire film is illuminated (cf. Figure 3), we expect vertical migration to still be prevalent and annihilation of defects under the illumination spot, leading to a net brightening of all grains, but the lateral migration will be dominated by local iodide and (filled) defect density gradients which will vary within and between grains; thus, some regions will locally brighten more than others due to relatively greater excess iodine removal in those regions."

5) To conclude, the paper contains interesting experimental observation, however, their interpretation is contradictory and unsatisfactory. I suggest reconsidering this paper in Nature Comm. if the authors:

1. Totally reconsider the interpretation of the TOF results.

We have now carefully considered the referee's concerns and added additional text and disclaimers addressing the ToF-SIMS results and a more thorough description of the results in the SI. Please see our response to point 1) above.

2. Reconsider the model, make it possible to explain photo-brightening of nano-sized crystals (Tain et al JPCLetters) and discuss the literature data where some models have been already presented (e.g. ref 14). Regardless with oxygen or without. Nitrogen-filled box used by the authors to prepare the sample does not totally exclude oxygen from the material. This is simply impossible.

We believe our model currently explains the effects in nano-sized crystals and we have now explicitly discussed this (see our response to points 2 and 3 above). Once again, this is the added paragraph:

"We believe that our mechanism is also consistent with the rapid photo-brightening in perovskite nanocrystals and nanocrystalline domains with few trap sites per crystal, as reported by Tian et al.⁴⁴ and Tachikawa et al.³¹. Here, traps are efficiently filled and the gradient driving ion migration results in the rapid net removal of the few interstitial-vacancy pairs per crystal."

We have also added in discussion of the Tian et al. model and how it relates to the model we propose (see response to point 2 above). We believe that they are compatible. Here, we have taken means to minimize oxygen content but we agree that we cannot guarantee absolute exclusion of oxygen. In any case, we believe the potential competition or even synergetic relationship between the photo-brightening from oxygen reaction or iodide diffusion will be an interesting question for the community moving forward. Once more, this is the added paragraph:

"The mechanism we propose here may not be able to explain all photo-brightening phenomena. For example, it has been recently shown that oxygen species could contribute to the photo-brightening process^{14,30}. Tian et al. recently proposed a model for the photo-brightening in the presence of oxygen, where photo-generated charge carriers and oxygen-related species interact to deactivate the trapping defects¹⁴. In their mechanism and in ours, the photo-brightening is limited by the interaction volume of photo-excited carriers in the film. We take means to minimize oxygen in our measurements presented here, suggesting that oxygen may not be essential in the process. Nevertheless, the competition between oxygen and iodine in both diffusion and ability to annihilate traps, and any possible synergy between the two, remains an ongoing question for the community."

We have also altered the text as follows:

"The extent of the PL enhancements is influenced by different atmospheres^{14,30,31}, though the effects are still observed in extremely low oxygen level environments including in vacuo and nitrogen and also in films made using other fabrication routes (see Supplementary Fig. 1), suggesting that the general photo-induced rise behaviour is intrinsic to these polycrystalline films."

3. Most probably it will not be possible for the authors to explain all observations in one model. However, admitting that the current status of the knowledge does not allow suggesting the definite mechanism of the photo-brightening in perovskites does not make the paper less interesting for the

community.

We agree with the referee and we believe that these sentiments are summarized in the text added for the previous point (above).

We have also made it clear that we cannot explain all phenomena when we introduce the model:

“Here, we propose a mechanism in Figure 6 that broadly describes our observations in this study, but it is likely that there are other complex mass transport mechanisms simultaneously occurring.”

Reviewer #2 (Remarks to the Author):

I have read the manuscript in its revised form and the response to reviewers submitted by the authors. The authors have convincingly answered to the major points raised by the reviewers and the manuscript is definitely improved, also including additional data on single crystals that confirm the original conclusions about light-induced defect migration. I believe the manuscript should now be published in Nature Comms. without further revisions.

We thank reviewer 2 for their time in reviewing our manuscript and suggesting that our manuscript be published in Nature Communications.

Extra remarks from reviewer #2

The reviewer argument could be reasonable, and it requires additional explanations from the authors.

If the TOF-SIMS signal is linearly dependent on total iodine concentration locally probed when scanning the position (Figure 4c), what the authors are measuring is a much larger variation than that expected based on the variation in trap density upon illumination. This latter quantity is correctly estimated by the reviewer to be less than $10^{17} / 10^{22} = 10^{-5}$, while the signal variation from Figure 4 is 1/10.

I am not saying the authors have misinterpreted the data here (there could be a volume factor that has not been included or explained or the signal may not be linearly dependent on concentration), but a further explanation is definitely in order. To make things clearer, I suggest to report data in Figure 4 (and 5) on a relative scale (i.e. % iodine enrichment) which is directly related to concentration, rather than the signal itself.

We would first like to direct reviewer 2 to our responses to reviewer 1, point 1. We believe we have addressed the major concerns and have also modified several sections of the text to clarify the information that we can extract from the ToF-SIMS experiment. As addressed in our response to reviewer 1, we are unable to report the data in an accurate way as % iodine enrichment. This is now discussed in the Supplementary Information. We believe that the most technically correct way is to still plot iodide intensity in terms of absolute counts.

Reviewer #3 (Remarks to the Author):

The manuscript was revised accordingly and is now suitable for publication without further revision.

We thank reviewer 3 for their time in reviewing our manuscript and suggesting that our manuscript be published in Nature Communications.

Reviewer #1 (Remarks to the Author):

In the revised version the authors did a good job in order to address my questions, explain their model and connect it to the ideas already existing in the literature. I still have some concerns regarding their data interpretation, however, I believe that the paper can be published in its present form. As the authors stated in the text, more research is needed in order to really understand what is going on upon light irradiation. However, the authors made definitely an important step forward which deserves publication in Nature Communication.